# SIRD: Transformers Assisted Step by Step Symbolic Integration

## Abstract

Recently, deep learning has gained popularity in solving statistical or approximate problems. However, working with symbolic data has been challenging for neural networks. Despite this, the natural sciences are making strides in utilizing deep learning for various use cases. In this work, we aim to solve the problem of symbolic integration by using deep learning through integral rule prediction, enabling faster search and better interpretability. We propose a novel symbolic integration rules dataset containing 27 million distinct functions and integration rule pairs. We show that by combining a transformer model trained on this dataset into SymPy's *integral_steps* function, the number of branches explored during the depth-first-search procedure was reduced by a factor of 3 and successfully solve functions that the original version was unable to handle.

## 1 Introduction

The advancements in computing have led deep learning to become more commonly used for many applications. Deep learning has been demonstrated to be highly effective in identifying statistical patterns. It delivers exceptional results in diverse areas such as computer vision (He et al., 2016; Dosovitskiy et al., 2020), speech recognition (Chan et al., 2016; Gulati et al., 2020), and natural language processing (NLP) Vaswani et al. (2017a); Devlin et al. (2018).

Deep learning for symbolic tasks, such as mathematical reasoning has not been extensively explored. Automating mathematical reasoning is one such area where numerous studies have been conducted, forming the foundation of these explorations. Notable studies include those by (Huang et al., 2023; Piotrowski et al., 2019; Zaremba et al., 2014; Loos et al., 2017).

Few attempts have been made to address the problem of symbolic integration using data-driven methods. Earlier works by Lample & Charton (2020) (hereafter referred to as 'LC'), Kubota et al. (2022), and Noorbakhsh et al. (2021) have laid a solid foundation in this area. However, there are not enough primary follow-up studies on the problem. LC proposed an end-to-end black-box model that frames the problem as sequence-to-sequence prediction. Using an encoder-decoder transformer, the model generates the integral given a mathematical function. Despite its good performance, this approach lacks explainability regarding whether the model memorizes specific patterns or can generalize the integration of complex functions by focusing on sub-expressions, as noted by Davis (2019).

Our approach to solving symbolic integration is similar to systems like AlphaGo (Silver et al., 2016) and AlphaGeometry (Trinh et al., 2024), where a neural model guides the steps toward solving a specific problem, such as chess in the case of AlphaGo and Olympiad geometry problems in the case of AlphaGeometry. These systems are efficient because they are trained on the steps of a specific task and are more interpretable than a system trained in a purely end-to-end fashion. Symbolic integration is essentially a search problem. When faced with a symbolic integration problem, people typically employ a systematic approach: they identify an applicable integration rule, apply the rule to obtain a new expression, and repeat the process until the final integral is found.

In this paper, we introduce the one-of-a-kind dataset Symbolic Integration Rules Dataset (SIRD-27M), consisting of nearly 27 million mathematical functions and their integration rules. This dataset is one step towards automating mathematical function integration in an interpretable manner using neural networks. We include the step-by-step integration rules for a given mathematical

function, similar to how we perform integration manually. At each step, we provide the integration rule that must be applied alongside the expression for complex rules like integration by-parts and u-substitution rules. Using the step-by-step integration procedure, we train a deep learning model that predicts the consequent rule and expression for a given mathematical function as the input. When performed repeatedly, this process simulates a human-like integration process. Compared to existing works that predict the integral directly in a black-box manner, this process is more accurate and interpretable. We also present robust benchmarks showing that SIRD can train a highly scalable and versatile model. Our approach could be applied to enhance the capabilities of general computer algebra systems.

The significant contributions of the paper are listed as follows:

- Introduced a large-scale, novel dataset - SIRD-27[1], that can be employed for performing interpretable step-by-step integration operation. The dataset contains over 27M mathematical functions, corresponding integration rules, and expressions for specific integration rules.

- We propose three mathematical integration tasks to gauge our models' performance for the symbolic integration task. These include - Complete Rule Prediction, Rule Prediction, and Integral Prediction.

- We used the symbolic maths package SymPy (Meurer et al., 2017) to perform integration given the model-predicted integration rules. To achieve this, we alter the *integral_steps* method of SymPy, replacing it with the proposed *guided_integral_steps* where we use the depth-first-search procedure to navigate the integration rules search space compared to *integral_steps* that depends on predefined static heuristics. Our proposed *guided_integral_steps* consistently explores $3\times$ fewer branches than *integral_steps*.

## 2 RELATED WORKS

Solving integrals using deep learning is part of the field of symbolic reasoning. Symbolic reasoning (Lavrac & Dzeroski, 1994; Newell & Simon, 2007) is a well-explored topic. The art of reasoning involves the intricate manipulation of symbols and the application of logical rules to perform deduction (Johnson-Laird, 1999), which involves drawing conclusions based on given premises, induction Lavrac & Dzeroski (1994), which involves making generalizations based on observed patterns, and abduction Kovács & Spens (2005), which involves generating hypotheses to explain observed phenomena. In visual question answering, (Yi et al., 2018) have introduced a technique that merges neural networks with symbolic rules. This integration enables the system to perform compositional and interpretable reasoning by leveraging visual and textual data. For the task of program synthesis, (Shin et al., 2018) suggested a method that incorporates inferred execution traces to guide the generation of accurate programs using LLMs.

Mathematical tasks such as multiplication (Kaiser & Sutskever, 2015; Zaremba et al., 2014), solving word problems (Huang et al., 2023; Wang et al., 2018; Chatterjee et al., 2022), and calculus (Lample & Charton, 2020; Panju & Ghodsi, 2020) have been attempted using deep learning. Despite progress, the current efforts remain largely black box, lacking interpretability and explainability.

In recent times, the community has been exploring the math-solving capabilities of large language models (LLMs), as highlighted in literature such as (Ji & Gao, 2023; Tang et al., 2023; Frieder et al., 2023). These models primarily focus on solving mathematical problems expressed in natural language involving numbers. However, calculus problems, expressed through various symbols and expressions without words, have been explored using a more end-to-end black-box method (Lample & Charton, 2020; Noorbakhsh et al., 2021), which calls for further exploration. In the past, various attempts (Rich et al.; Meurer et al., 2017) have been made to develop parsing rules for different mathematical functions and use calculus rules to solve problems intuitively. Calculus problems are search problems where the final solution is several mathematical steps away.

One of the closest works by LC used transformers for symbolic integration. This work utilized language models, notably a vanilla transformer, for training with differential equations and integration problems as input to the network. The model was found to memorize specific functional patterns

---

[1]Dataset available at `http://tiny.cc/sird27m`. Supplementary Material for code and model weights

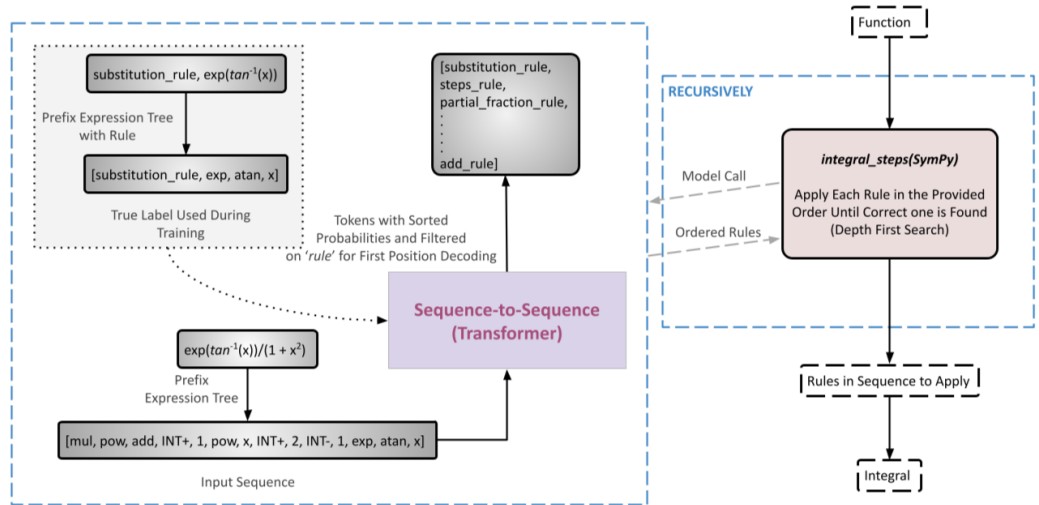

Figure 1: End-to-end integral prediction using *guided_integral_steps*

and provided correct integrals to functions where even state-of-the-art symbolic mathematical computation programs such as Mathematica (Wolfram Research Inc.) failed. However, as pointed out by Davis (2019), the model has no innate knowledge of any of the integration rules, and its success is mainly due to repetitive exposure to a certain pattern of mathematical functions.

## 3    SIRD-27M: Symbolic Integration Rules Dataset

We train an encoder-decoder model on integral calculation steps rather than integrals (more in Section 4.1). For this, we developed a Symbolic Integration Rules Dataset (SIRD-27M) that consists of 27 million function-integration rule pairs. These pairs represent occurrences in the integral calculation steps when solving integrals for various functions. This dataset encompasses a total of 24 different integration rules (see Appendix C for details), including complex rules such as the *substitution rule* and *integration by parts*. Since applying these complex rules also requires an expression, we provide the expression to be used along with the rule. The following sections discuss the steps of generation, validation, and processing for SIRD-27M.

### 3.1    Data Source

LC introduced three datasets for symbolic integration: the Forward Method (FWD) Dataset, the Backward Method (BWD) Dataset, and the Integration by Parts (IBP) Dataset, released under CC BY-NC 4.0 license (Lample & Charton, 2020). All three datasets consist of function and integral pairs, their difference lies in how the samples are generated. Hence, relatively, each of the three is out-of-distribution(OOD) with respect to the others. This variance affects parameters such as the length of function and integral expressions. The characteristics of each dataset are as follows:

**Forward Method (FWD) Dataset**: In this dataset, the authors generate random functions and perform integration using the *integrate* method from SymPy (see Appendix A.1 for details). The dataset is named for its forward generation direction, moving from the function to the integral.

**Backward Method (BWD) Dataset**: Here, authors generate random functions representing the integrals. These functions are then differentiated to obtain the functions corresponding to the integrals. The generation direction is backward, moving from the integral to the function through differentiation, hence the name.

Table 1: SIRD examples.

| Function | Rule | Application Expression | Model Input Sequence | Model Output Sequence |
|---|---|---|---|---|
| $x^2 + 2x$ | add_rule | - | [add, pow, $x$, INT+, 2, mul, INT+, 2, $x$] | [add_rule] |
| $\exp(tan^{-1}(x))/(1+x^2)$ | substitution_rule | $\exp(tan^{-1}(x))$ | [mul, pow, add, INT+, 1, pow, $x$, INT+, 2, INT-, 1, exp, atan, $x$] | [substitution_rule, exp, $atan$, $x$] |

**Integration by Parts (IBP) Dataset**: This dataset used the integration by parts integration rule to generate the function-integration pair. For two randomly generated functions $u$ and $v$:

$$\int u\,dv = uv - \int v\,du \tag{1}$$

There are two integral terms in equation 1. The term $\int v\,du$ is already present in the generated data and $\int u\,dv$ is calculated using a simple subtraction operation with $uv$, thus giving rise to a new sample.

In this work, we have utilized functions from the FWD dataset to generate function-integration rule pairs for SIRD-27M. To generate the intermediary function-integration rule pairs while solving for an integral, we modify the *integral_steps* method from SymPy. See Appendix A for more details on the mentioned SymPy methods and modified *integral_steps* to generate SIRD samples. We selected FWD dataset because its samples are more likely to be successfully integrated by *integral_steps* due to an integration operation being used in its generation as described above while BWD is generated by differentiating random functions. As a result, integrating BWD samples directly is often more challenging and prone to errors, such as recursion stack overflow. In contrast, the FWD dataset enabled the generation of more function-integration rule pairs for SIRD.

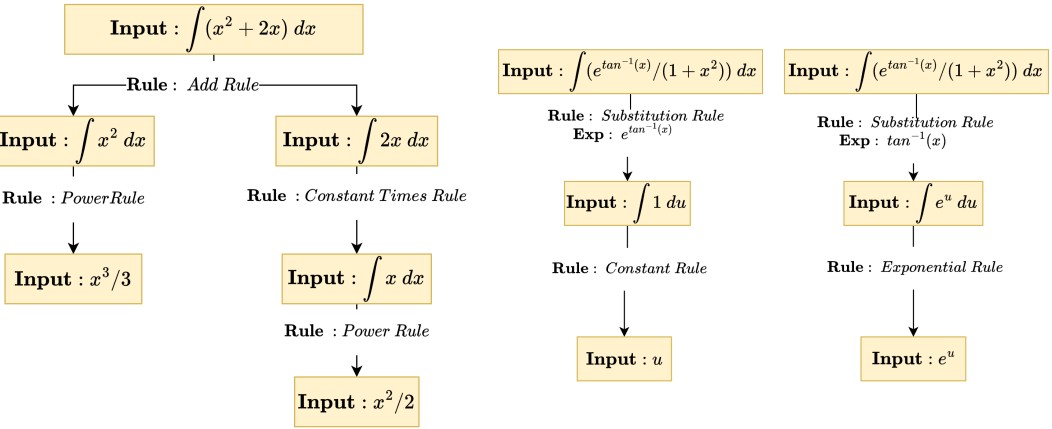

Figure 2: A tree showing the rules the model chooses while performing a search. (*left*) $x^2 + 2x$ (*right*) $\frac{\exp(tan^{-1}(x))}{1+x^2}$

## 3.2 DATA GENERATION

We extract the functions and integration rules at each step when solving functions from the FWD dataset. One step, i.e., a pair of intermediary function and integration rule, is used as a sample in SIRD. We modify individual integration rule methods provided by SymPy and the flow of the *integral_steps* method to create a data generation script that outputs all the steps (function and integration rule pairs) for a function from FWD. Appendix A.2 provides examples and a detailed explanation of the output syntax and step-by-step data generation script.

Table 1 shows a few examples from SIRD-27M. The third column, *Application Expression*, contains expressions that might be required for the application of particular integration rules, such as the *substitution rule* or *integration by parts rule*. A mathematical function can be integrated through

various routes of integration rules when a person follows the typical step-by-step process to solve an integration problem. As we are modelling integration using intermediary steps, it is possible that in SIRD-27M, we have different integration rules corresponding to the same function. The two trees on the right in Figure 2 give an example where the same function can be integrated in multiple ways. The tree on the left shows a typical step-by-step integration with intermediary functions and rules.

## 3.3 DATA VALIDATION

The *integral_steps* method, in its original form, outputs a sequence of integration rules that can be used to calculate the integral of a function. Our data generation script not only produces this sequence of integration rules but also generates explicit pairs of function-integration rules for each step. To validate the correctness of our data generation script, we employ two checks:

- First, we directly compare the output of the original *integral_steps* method with the output of *integral_steps* generated by our data generation script for an input mathematical function.
- Second, we apply the output sequence of integration rules from our data generation script to obtain the final integral of the input mathematical function and differentiate it. Then, we compare the derivative with the input mathematical function.

This process ensures the correctness of the intermediary functions and integration rules generated by our data generation script.

## 3.4 DATA PROCESSING

After generating the intermediary steps, we segregate all the function and integration rule pairs, each forming a sample in SIRD. Hence, a single function from the FWD dataset leads to multiple samples for SIRD - one for each step towards solving the integral of the function. Afterwards, we remove the duplicate entries in the dataset.

There are two types of integration rules in the dataset based on their application behaviour: *a)* simple rules such as the *add rule*, *multiplication rule*, *power rule*, etc., which do not require any additional expression for their application (refer to the *left* in Figure 2) *b)* complex rules such as the *substitution rule*, *integration by parts*, etc., which require an additional expression for their application (refer to the *right* in Figure 2). For complex rules, the application expression is also available in SIRD.

Looking at a sample from SIRD for training, a mathematical function serves as the input to the model, and an integration rule along the application expression, if present, serves as the label. It is required to convert a mathematical function (model input) into a sequence of tokens that maintains the relationship between symbols, numbers, and operators for training a sequence-to-sequence model. Expression trees are widely used to represent mathematical functions. These trees can be transformed into sequences of tokens using prefix, infix, or postfix notations. Similar to LC, we adopt the prefix notation of the expression tree, wherein a node is positioned before its children. This notation offers the added advantage of not necessitating parentheses, as required in infix notation, resulting in a shorter sequence length when representing the mathematical function.

As for the label, if the sample has a simple rule (without an application expression), there will be a single-token output sequence, i.e., the integration rule name. In the case of a complex rule (with an application expression), the application expression is also converted to prefix notation and appended after the rule name token. This approach to converting the label into a sequence helps standardize how labels are utilized for model training. It simplifies the process for the model to learn representations for three key decisions: determining which rule to apply, whether there will be an application expression, and identifying the content of the application expression. For examples of input and output sequences, please refer to the fourth and fifth columns in Table 1. Additionally, the distribution of input sequence lengths is presented in Appendix B.

We have used same strategy for tokenisation as LC. Our tokens encompass various unary operators (such as sin, cos, tan, exp, etc.), binary operators (like add, sub, mul, pow, etc.), specific symbols (e.g., x, u, y, etc.), and numerical values. Furthermore, we've included 24 additional tokens corresponding to the integration rules. As previously mentioned, these tokens facilitate the creation of an expression tree for the input mathematical function and its corresponding prefix notation. Few token examples can be seen in Table 1

## 4 EXPERIMENTS

### 4.1 MODEL ARCHITECTURE

We have formulated a step-by-step modelling approach. Given a mathematical function as input, we train a model to learn which integration rule to apply next to compute its integral. This is modelled as a sequence-to-sequence prediction. where in given a single variable mathematical function $F_{expr}$, corresponding integration rule and expression (if required in the application of integration rule) $G_{rule\&expr}$, an encoder-decoder model is trained on the following objective:

$$\min \frac{1}{n} \sum_{i=1}^{n} l(S_{\Theta}(F_{expr}), G_{rule\&expr}) \tag{2}$$

where $l$ denotes loss function used to train the model, $S_{\Theta}$ represents the encoder-decoder model, $\Theta$ represents the learnable model parameters and $n$ denotes number of training samples.

We use SIRD-27M to train the model. We utilize the Transformer architecture from Vaswani et al. (2017b) for the sequence-to-sequence model. Besides symbolic integration, LC empirically found that the Transformer architecture performs relatively well for a few other symbolic mathematics tasks. Hence, we decided to maintain the same architecture and configuration.

Regarding the configuration of the Transformer architecture, we employ eight attention heads, six layers, and an embedding size of 512. We utilized the Adam optimizer Kingma & Ba (2014) with a learning rate of $4 * 10^{-5}$. The model is trained using a batch size of 256 samples. We limited the number of tokens in input expressions to 384 and used approximately 21 million samples (80% of SIRD-27M) for training. The following sections will discuss the definitions, details and evaluations of different benchmark tasks.

Table 2: Accuracy results for Rule Prediction Tasks. Evaluation is done on 10% SIRD-27M

| Task | Test Set | Number of Samples | Approach | Additional Granularity | Accuracy (%) |
|---|---|---|---|---|---|
| Complete Rule Prediction | SIRD-27M | 2742469 | Our model | – | 80.41 |
| Rule Prediction | SIRD-27M | 2742469 | Our model | – | 81.44 |
| | | 873941 | | Add Rule | 99.96 |
| | | 734199 | | Multiplication Rule | 98.34 |
| | | 191826 | | Substitution Rule | 52.74 |
| | | 94897 | | Parts Rule | 65.74 |
| | | 847606 | | Other Rules | 55.97 |

### 4.2 RULE PREDICTION TASKS

We have evaluated our model trained on SIRD on following integration rule prediction tasks.

**Complete Rule Prediction:** This task aims to observe if, along with the integration rule, the model can learn which rule should be accompanied by an application expression and predict it accurately. We evaluate this task on a randomly selected 10% holdout test set of SIRD-27M. A prediction is correct if the predicted integration rule and application expression match the ground truth label. For this task, we keep decoding using the model until the end-of-sentence token is observed.

**Rule Prediction:** In this task, we assess whether the model can accurately predict the correct integration rule given a mathematical function. Here, we do not consider the application expression in the evaluation. Here, evaluation is also done on a randomly selected 10% holdout test set of SIRD-27M. A prediction is correct if the predicted integration rule matches the ground truth label. For this task, we decode for the single token as the rule token is the first in the output sequence during model training (Section 3.4).

Table 2 shows the accuracy results for the above-described tasks. We have also quoted individual rule accuracy numbers for the Rule Prediction task along with the overall accuracy. While reading these results, it is essential to note that there may be multiple correct rules to apply for a given mathematical function for the rule prediction tasks (as there can be more than one way to solve integral). However, in the test set, we only have a single ground truth label for the function.

Table 3: Results for **Integral Prediction task**. LC model trained on FWD is used. Beam size 1 is used for decoding the LC model. Timeout of 45 min per function is used for all the approaches.

| Test Set | Number of Samples | Approach | Accuracy (%) |
|---|---|---|---|
| FWD | 7000 | integral_steps | 95.55 |
| | | guided_integral_steps | **95.84** |
| | | LC | 93.39 |
| BWD | 4400 | integral_steps | **35.09** |
| | | guided_integral_steps | 34.00 |
| | | LC | 21.17 |
| IBP | 5700 | integral_steps | 92.06 |
| | | guided_integral_steps | **93.07** |
| | | LC | 89.28 |

Table 4: Accuracy Results with limit on number of explored nodes for **Integral Prediction task**

| Test Set | Number of Samples | Node Limit | integral_steps | guided_integral_steps |
|---|---|---|---|---|
| FWD | 7000 | 100 | 79.85 | 93.71 |
| | | 80 | 73.91 | 92.65 |
| | | 60 | 66.48 | 90.25 |
| | | 40 | 55.74 | 84.37 |
| BWD | 4400 | 400 | 11.66 | **19.48** |
| | | 250 | 10.43 | 13.84 |
| | | 200 | 10.04 | 11.87 |
| | | 180 | 9.85 | 11.76 |
| IBP | 5700 | 100 | 84.69 | 90.5 |
| | | 80 | 80.26 | 89.81 |
| | | 60 | 76.44 | 87.97 |
| | | 40 | 71.26 | 83.01 |

## 4.3 INTEGRAL PREDICTION

**Integral Prediction:** This task evaluates the end-to-end performance of the model in calculating integrals. To achieve this, we developed the *guided_integral_steps* method (detailed below) by integrating our model with SymPy's *integral_steps* method. *guided_integral_steps* applies integration rules based on our model's predictions at each step. A prediction is correct if the derivative of the resulting integral matches the input function.

We have reported results using the test sets from the FWD, BWD, and IBP datasets. The function lengths and complexities in the SIRD-27 dataset are simpler, as they represent intermediary steps, making the other three datasets out-of-distribution (OOD) and ideal for demonstrating the generalization of our approach. Furthermore, considering that SIRD-27M is based on samples from FWD, we also included evaluations on the BWD and IBP datasets. Both datasets significantly differ from FWD in generation method and function characteristics, as described in Section 3.1. However, the current version of SIRD does not support complex mathematical functions involving hyperbolic trigonometric operators and their inverses (e.g., *sinh(x)*, *cosh(x)*). The *integral_steps* method cannot handle such functions. Consequently, we exclude such functions from the test sets of all three datasets before evaluation. Additionally, we remove functions with a sequence length greater than 384 when converted to prefix notation.

*Guided_integral_steps*: It is created by incorporating our model into the *integral_steps* method of SymPy. In its original implementation, *integral_steps* calls SymPy methods corresponding to each integration rule in a fixed order defined heuristically. The integration rules are checked at every intermediate step to determine their applicability until the correct one is identified. It performs a depth-first search through all the defined integration rules at each intermediary step when solving for the integral.

In the *guided_integral_steps*, we removed the fixed order of integration rules defined heuristically. Instead, we run an inference through our model with the expression of the current intermediary step as input. It performs decoding only for a single token and outputs the probability for each integration rule token, based on which these rules are ranked and tried at each step. This way, the order of integration rules explored during the depth-first search becomes dynamic as it depends on the expression of the current intermediary step. Figure 1 can be referenced for the visual representation of this flow.

Table 5: Efficiency based on average nodes explored per test function

| Test Set | integral_steps | guided_integral_steps | Efficiency |
|---|---|---|---|
| FWD | 92.98 | 25.25 | **3.7×** |
| BWD | 1187.37 | 380.39 | **3.12×** |
| IBP | 49.33 | 17.13 | **3×** |

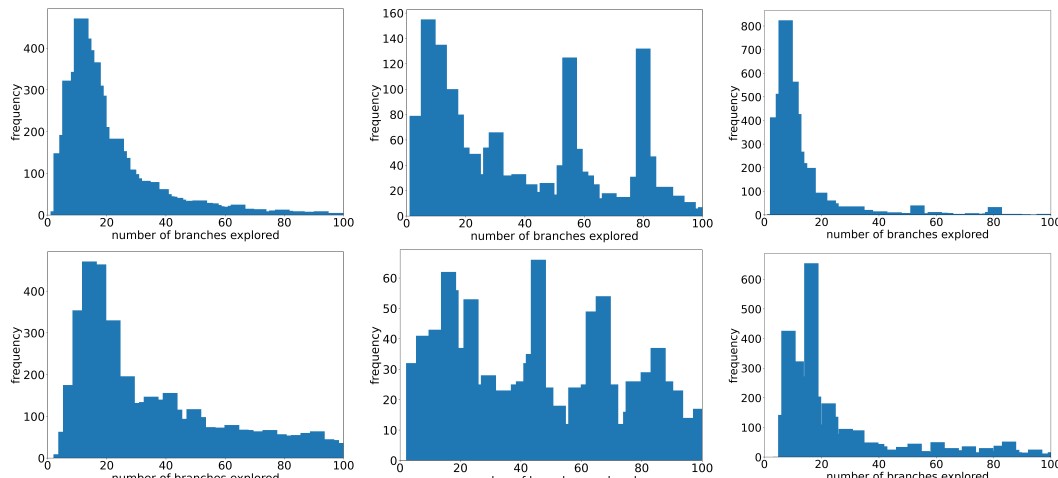

Figure 3: Branches explored during depth-first-search using *(top) guided_integral_steps (bottom) integral_steps*. [*left to right*] *(first)* FWD test set *(second)* BWD test set *(third)* IBP test set

**Accuracy Comparison:** Table 3 shows that *guided_integral_steps* outperforms the LC model trained on FWD across all three test sets. Notably, *integral_steps* also outperforms the LC model, but *guided_integral_steps* achieves a higher success rate than *integral_steps* for the FWD and IBP test sets. However, for the BWD test set, both approaches perform similarly. The larger function length in BWD results in longer absolute times for *guided_integral_steps* due to the model inference call overhead for each step. In the other two test sets, efficiency improvements (discussed in the next paragraph) offset this time overhead, which is not the case for BWD. A timeout of 45 minutes was set for individual functions in these results.

We have ensured the experiments are fair by selecting LC's model trained on the FWD dataset. LC showed that models trained on FWD, BWD, or IBP perform well on their own test sets but struggle on others, revealing an out-of-distribution issue. While training on all datasets combined improves cross-test performance (Lample & Charton, 2020), this would make the comparison with *guided_integral_steps* unfair. Additionally, it's unclear whether the improvement in cross-test performance reflects true generalization or pattern memorization, as Welleck et al. (2021) pointed out. Comparing to LC's FWD model helps demonstrate the OOD performance of *guided_integral_steps* using the BWD and IBP test sets.

**Efficiency Comparison:** Along with having a better overall success rate, *guided_integral_steps* is highly efficient, i.e. given a limit on a number of nodes that can be explored at each step *guided_integral_steps* highly outperforms *integral_steps*. In Table 4, we show that the success rate for *guided_integral_steps* is far better across test sets for different node limits compared to *integral_steps*.

Regarding average nodes explored by each sample, Table 5 shows the efficiency of *guided_integral_steps* across the three sets. Not only FWD, which is used to create SIRD-27M, but it was also able to optimize node exploration for BWD and IBP, demonstrating a highly generalizing nature. Along with the average results Figure 3 shows the distribution of nodes explored by the samples, and left skewness can be seen in the top row, which further showcases that *guided_integral_steps* can solve integral with the need of exploring a far lesser number of nodes.

We also measured the average runtime for the functions in the FWD test set, with *guided_integral_steps* averaging 0.34 seconds compared to 0.52 seconds for *integral_steps*. Some functions are significantly more complex and take minutes to complete, where the speedup is also evident. For practical application in a production system, inference overhead can be further reduced by techniques such as caching results for commonly occurring expressions, thereby minimizing the number of inference calls. Additionally, batching multiple inference calls can be employed to optimize performance.

## 5 CONCLUSION

We present a novel step-by-step approach to the symbolic integration problem. We frame integration as a search problem and accelerate it using AI, resulting in a fast, accurate, and interpretable approach. We introduce a new dataset called SIRD-27M, where the task is to predict the integration rule that should be applied to a given function to find its integral. We show that a model trained on SIRD can guide the search for the integral, outperforming heuristics-based search and showing superior generalization ability. Our work is a preliminary but strong exploration of using deep learning for step-by-step symbolic integration, leading the way for further research.

With this work, we are laying the primary foundation for AI-assisted problem-solving in the symbolic integration domain, and it would be intriguing to explore whether its use cases can be extended to other scientific problems.

## 6 LIMITATIONS & FUTURE DIRECTIONS

Though our approach performs well in guiding the search for integral rules, one of the primary drawbacks of this approach is that it depends on the underlying step-by-step integration algorithm *integral_steps* in our case. Furthermore, there is an overhead of model inference calls at each step, and inference optimization might be required to optimize the runtime performance of *guided_integral_steps*.

The search methodology can be further refined with a beam search encompassing all intermediary steps when solving integrals for a function. This enhancement will further improve guided integral steps. Additionally, in this version of our work, we do not utilize the application expressions predicted by the model for complex rules; however, incorporating them could dramatically reduce the time required to solve integrals.

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

# A SYMPY FUNCTIONS & SIRD SAMPLE GENERATION SCRIPT

## A.1 FUNCTIONS FROM SYMPY

For the generation of FWD, Lample & Charton (2020) (hereafter referred to as 'LC') used *integrate* function from SymPy, and we used the modified version of *integral_steps* function to generate samples for SIRD utilizing functions from FWD. Here, we briefly explain both the functions and on what grounds these differ.

- ***Integrate*** Function: This is the principal method to integrate functions in SymPy. This function uses the Risch-Norman algorithm (Risch, 1969) and can solve both definite and indefinite integrals, though it does not provide the steps of integration.

- ***Integral_steps*** Function:
    - This function imitates how a human would solve an integral problem step by step. It outputs all the intermediate steps (expressions and integration rules) required to solve the integral of a given function. Further, these intermediate steps can be input to another function *__manualintegrate__*, which can apply the steps to generate the final integral of the function.
    - To calculate output of *integral_steps*, various integration rules' are implemented in SymPy. Whenever called with an input, it heuristically searches which integral rule to apply based on a defined order in the codebase. There are certain functions such as the *add_rule* which recursively again calls *integral_steps* to solve the subproblems. It tries all the potential rules in a predefined order to either output steps to solve the integration of a function or fallback to *DontKnowRule*.

Based on the description, it is clear that *integrate* function is far more capable of solving an end-to-end integration problem than *integral_steps*. However, we rely on the latter for our study as we require integration rules instead of just the final integral for a function. Also, currently, *integral_steps* cannot generate integration rules for complex functions involving hyperbolic trigonometric functions.

Given the nature of *integral_steps* implementation for many functions, it might go into an infinite search loop which either can be broken by *timeouts* or *Maximum Call Stack Exceeded*.

## A.2 MODIFIED *integral_steps* TO GENERATE SIRD SAMPLES

As described in section A.1, *integral_steps* function generates all the intermediary steps' expressions and integration rules while solving for a function using heuristically searching for correct rules to apply. However, it generates its output in particular syntax, which can be highly dynamic for different types of inputs. Following are the examples of its output:

---

**Function:** $x^2 + 2x$
**Output from *integral_steps*:** *AddRule(substeps=[PowerRule(base=x, exp=2, context=x\*\*2, symbol=x), ConstantTimesRule(constant=2, other=x, substep=PowerRule(base=x, exp=1, context=x, symbol=x), context=2\*x, symbol=x)], context=x\*\*2 + 2\*x, symbol=x)*

---

**Function:** $e^{tan^{-1}(x)}/(1 + x^2)$
**Output from *integral_steps*:** *AlternativeRule(alternatives=[URule(u_var=_u, u_func=exp(atan(x)), constant=1, substep=ConstantRule(constant=1, context=1, symbol=_u), context=exp(atan(x))/(x\*\*2 + 1), symbol=x), URule(u_var=_u, u_func=atan(x), constant=1, substep=ExpRule(base=E, exp=_u, context=exp(_u), symbol=_u), context=exp(atan(x))/(x\*\*2 + 1), symbol=x)], context=exp(atan(x))/(x\*\*2 + 1), symbol=x)*

---

Writing parsing rules for the above output can make things unnecessarily complex, and there can be many exceptions, given the highly dynamic nature of output syntax. Hence, to get exact

subexpression-integration rule pairs for a function, we created a data generation script by doing the following:

- Modified each integration rule function in SymPy to output a tuple of subexpression and rule name.
- For rules like *substitution_rule*, which require transforming a subexpression of a function for application, we also included the subexpression to be transformed along with the rule name in the output tuple.
- Modified the flow of *integral_steps* to accommodate this extra output along with the original.

Following are examples of output from our data generation script for the same functions:

---

**Function:** $x^2 + 2x$
**Output from Our Script:** *(AddRule(substeps=[PowerRule(base=x, exp=2, context=x\*\*2, symbol=x), ConstantTimesRule(constant=2, other=x, substep=PowerRule(base=x, exp=1, context=x, symbol=x), context=2\*x, symbol=x)], context=x\*\*2 + 2\*x, symbol=x),*
*[(IntegralInfo(integrand=**x\*\*2 + 2\*x**, symbol=x), **'add_rule'**),*
*(IntegralInfo(integrand=**x\*\*2**, symbol=x), **'power_rule'**),*
*IntegralInfo(integrand=**2\*x**, symbol=x), **'mul_rule'**,*
*(IntegralInfo(integrand=**x**, symbol=x), **'power_rule'**)]).*

---

**Function:** $e^{tan^{-1}(x)}/(1 + x^2)$
**Output from Our Script:** *(AlternativeRule(alternatives=[URule(u_var=_u, u_func=exp(atan(x)), constant=1, substep=ConstantRule(constant=1, context=1, symbol=_u), context=exp(atan(x))/(x\*\*2 + 1), symbol=x), URule(u_var=_u, u_func=atan(x), constant=1, substep=ExpRule(base=E, exp=_u, context=exp(_u), symbol=_u), context=exp(atan(x))/(x\*\*2 + 1), symbol=x)], context=exp(atan(x))/(x\*\*2 + 1), symbol=x),*
*[(IntegralInfo(integrand=**exp(atan(x))/(x\*\*2 + 1)**, symbol=x), **'substitution_rule', exp(atan(x))**)),*
*(IntegralInfo(integrand=**1**, symbol=_u), **'constant_rule'**),*
*(IntegralInfo(integrand=**exp(atan(x))/(x\*\*2 + 1)**, symbol=x), **'substitution_rule', atan(x)**),*
*(IntegralInfo(integrand=**exp(_u)**, symbol=_u), **'exp_rule'**)]).*

---

This way, it becomes straightforward to parse the expression-integration rule pairs for a function constituting SIRD samples.

## B  INPUT EXPRESSION LENGTH

Before training a sequence-to-sequence model on SIRD-27M, we filtered the training data samples by the following criteria: The input expression prefix form should not exceed 384 tokens in length, and the output sequence, which includes both the rule name and the accompanying expression for eligible rules, should be no longer than 29 tokens. Figure 4 displays the distribution of input sequence lengths for all samples in SIRD-27M.

## C  SIRD-27M: DATASET STATISTICS

Symbolic Integration Rules Dataset (SIRD) consists of 27 million+ samples, including 24 integration rules. Table 6 lists the frequency of different integration rules in SIRD.

Certain rules have a rather small number of samples. However, we would like to point out that this is just the representation of patterns/steps we extracted from 10 million samples of the FWD dataset. When integrating an unseen function, the add rule will still be used ten times more than other obscure rules. Since the distribution of the rules in our training set matches the distribution in our testing set, this imbalance is not an issue but a strength in reality.

Particularly for *exp_rule*: Our dataset consists of distinct integral steps-rules pairs rather than function-integrals pairs. Most of the complex functions involving exponential terms for example $e^{x^2}$ will go through either the substitution rule or by parts rule. Only after which exponential rule

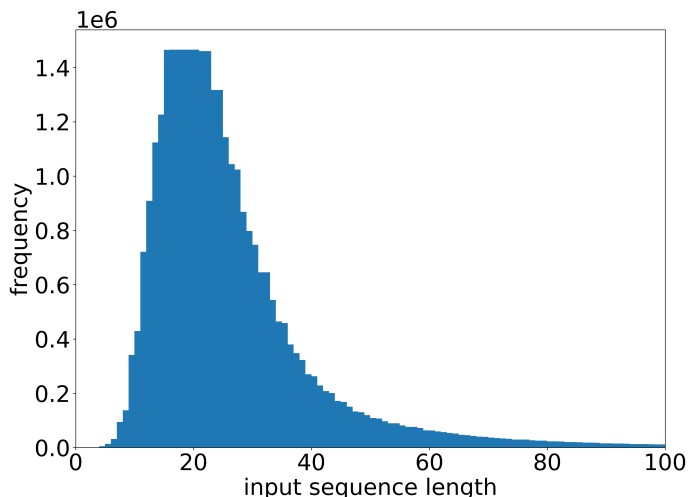

Figure 4: The histogram of the sequence lengths for functions in SIRD-27M.

Table 6: Integration rules and their frequencies in SIRD-27M.

| Rule Name | No. of Samples |
| --- | --- |
| add_rule | 8768793 |
| mul_rule | 7348761 |
| partial_fractions_rule | 3014738 |
| substitution_rule | 1922493 |
| cancel_rule | 2284315 |
| distribute_expand_rule | 1935582 |
| parts_rule | 951144 |
| sqrt_linear_rule | 332918 |
| quadratic_denom_rule | 388780 |
| constant_rule | 324179 |
| trig_rule | 27561 |
| trig_expand_rule | 123965 |
| sqrt_quadratic_rule | 31261 |
| trig_sincos_rule | 3674 |
| inverse_trig_rule | 1632 |
| power_rule | 2547 |
| trig_sindouble_rule | 1810 |
| trig_tansec_rule | 755 |
| special_function_rule | 98 |
| trig_cotcsc_rule | 39 |
| trig_substitution_rule | 113 |
| hyperbolic_rule | 6 |
| exp_rule | 2 |
| trig_product_rule | 2 |

can be applied. So in this case we will substitute $x^2$ and then will apply an exponential rule on it. So it is evident that when looking at the steps we will end up getting the same function-rule pair with possibly different variables for exponential rule. Hence 2 samples in our dataset represent simple exponential functions with different variables.

For *hyperbolic_rule*, as mentioned in Line 368 of Section 4.3, a limitation of SymPy is that its *integral_steps* method can not handle complex functions involving hyperbolic trigonometric operators. Hence, in our dataset, six samples represent two samples with different variables each for cosh, sinh and tanh functions.

## D  GENERALIZATION BEYOND THE GENERATOR - SYMPY *integral_steps*

We have observed that *guided_integral_steps* can integrate certain functions that *integral_steps* cannot. We have demonstrated few such examples in Table 7.

Table 7: Examples where the original *integral_steps* fails but *guided_integral_steps* succeeds.

| **Function** |
| --- |
| $e^x + (\cos^{-1}(x))^2$ |
| $x + (\sin^{-1}(x))^2$ |
| $\sin(\sqrt{x}\tan(5))$ |
| $(\cos^{-1}(x))^2 + \cos^{-1}(x) + \frac{1}{4}$ |
| $4x^2 + 4x\sin^{-1}(x) - 20x + (\sin^{-1}(x))^2 - 10\sin^{-1}(x) + 25$ |
| $2\sqrt{x}e^{\sqrt{x}} + e^{2\sqrt{x}}$ |

