# OpenReview forum: "SIRD: Transformers Assisted Step by Step Symbolic Integration"
_ICLR.cc/2025/Conference — Submitted to ICLR 2025_

### Official Review · Reviewer_qgw3 · 2024-10-28

**Soundness:** 3
**Presentation:** 1
**Contribution:** 2
**Rating:** 3
**Confidence:** 2

**Summary:**

Previous work on automated symbolic integration has framed the task as a sequence-to-sequence modeling problem, aiming to directly predict the result of integration without intermediate steps. In this work, the authors approach automated symbolic integration as a program search problem over the space of integration rules. Rather than inducing the integral of a mathematical function directly, the authors employ a neural-guided search strategy. A trained transformer model selects an appropriate integration rule based on the current state, which is then executed by a symbolic mathematics engine (SymPy) to generate the next state. To train the neural guidance model, the authors introduce a dataset containing 27 million pairs of procedurally generated mathematical functions and their corresponding integration rules. When compared to a symbolic search method without neural guidance (SymPy), the proposed approach is more efficient (in the number of branches explored). On a held out dataset of problems, the proposed method slightly outperforms the black-box sequence-to-sequence algorithm.

**Strengths:**

- Automated symbolic integration is a well-established and widely recognized problem within the ML community. This problem is highly significant and relevant to the ICLR audience.
- Previous work in this area `[1]` generated a lot of interest and the community found a lot of avenues of improvement. Many of the avenues of improvement were summarized in this review `[5]`. A major limitation identified in `[1]` was the behavior of the fully neural model when given a challenging integration problem. A robust integration system should ideally give up on unsolvable cases; however, the sequence-to-sequence model in `[1]` often produced a confident but incorrect answer.
    - The current approach circumvents this hallucination problem by adopting a hybrid solution, where the neural network output is restricted to be within the symbolic search space of SciPy's integration rules. This design allows the algorithm to fail gracefully when no solution is found, while still benefiting from the accelerated search capabilities offered by the neural model. This is pretty neat!

**Weaknesses:**

_All relevant weaknesses are in the “Questions” section. I summarize my reasoning for the current score here. I am open to discussing these points with the authors and potentially revising my recommendation during the discussion period._

Overall, I'm recommending this paper for **Rejection**. While SIRD presents an exciting approach to automated symbolic integration, the current scope of the training and evaluation dataset is too narrow to form reliable observations about the overall efficacy of this work for general-purpose symbolic integration. Furthermore, the presentation is severely lacking and will benefit from another round of reviews.

**Questions:**

* **Dataset generation concerns:**
* For procedurally generated datasets, the diversity of questions and the relative difficulty are more critical than the scale of the dataset.
  * Previous work in this area `[1]` generated a lot of interest and the community found a lot of avenues of improvement. Many of the avenues of improvement were summarized in this review `[5]`. Specifically, a key problem with the evaluation of `[1]` was that the comparison with SymPy, Mathematica, and Matlab only targeted a very small fraction of in-domain problems (verbatim: "indefinite integrals of enormously complex functions of a single variable ‘x’ whose integral is a much smaller elementary function containing no constant symbols other than the integers −5 to 5.").
* Does the current procedural generation script expand the diversity of the problem set beyond these narrow constraints?
* **Additional Evaluation Dataset**:
  * The held-out dataset may not be a sufficiently challenging benchmark for general automated symbolic integration algorithms, given that both the proposed algorithm and `[1]` achieve over 90% accuracy. A more demanding evaluation could enhance the robustness of the results. Here's are some suggestions for "gold standard" datasets to evaluate the efficacy of this method:
    * The MIT integration bee is an annual competition in which contestants are asked to integrate an equation in the least amount of time. The practice rounds consists of 20 questions. All the previous year questions (and answers) are available at the bottom of this website: https://math.mit.edu/~yyao1/integrationbee.html
    * Are current SOTA methods like SymPy, `[1]`, and the proposed algorithm able to solve some of these problems? Although some mathematical operators (e.g., hyperbolic functions like `tanh`) may be out-of-domain, a significant portion of these problems should still fall within the scope of this approach.
    * The proposed algorithm should ideally perform as well as SymPy, whereas the baseline `[1]` is likely to struggle on these more complex, out-of-domain problems.
    * I want to emphasize that the results on this evaluation should not be held against the contributions in this paper. Most of these problems are very challenging for humans without prior training. This task is not a standalone benchmark for general symbolic integration capabilities, but may serve as a valuable test of the algorithm’s overall efficacy.
  * Of course, the authors may prefer other gold-standard datasets for downstream performance evaluation.
* **Additional Evaluation Baseline**:
  * Symbolic integration necessitates high level planning and reasoning abilities. As such, a comparison with an LLM based baselines would substantially strengthen the paper’s central claims. I suggest evaluating against an instruction-finetuned LLM in a zero-shot program induction settings. ie: Directly prompt an LLM with the integrand and instruct it to derive the solution. A prodecurally generated dataset should ideally not cause any data leakage issues.
* **Clarity:**
  * I apologize if this sounds harsh, but the paper is poorly organized. A research paper should generally present a series of properly organized logical statements.  However, the current manuscript scatters key information across many sections. I highly recommend reviewing `[2,3,4]`. Here is a (incomplete) list of clarity issues:
    * The introduction doesn't summarize key results. In the bullet points that summarize contributions, there is no mention of a comparison with `[1]`. This was very confusing on the first pass, as I thought the authors only evaluated against SymPy. It isn't until Page 6 that we see a comparison with `[1]`.
    * Why is the loss formulation in the experiments section? Consider creating a problem statement section that defines the search problem, the neural guided search problem, the loss formulation for the search problem, and the search space.
      * The preliminaries section of these program synthesis papers might be helpful.
        * https://arxiv.org/abs/1906.04604
        * https://arxiv.org/pdf/2006.08381
        * https://arxiv.org/abs/2310.19791
        * https://arxiv.org/abs/2007.12101
        * https://arxiv.org/abs/2409.09359
    * Figures are poorly motivated, and the captions lack depth. I suggest the authors rewrite the captions with a brief description of the experiment, the hypothesis, the relevant findings, and how the relevant findings support the central claim of the experiment.
    * The explanation of integration by parts might be unnecessary within the section on dataset generation. I recommend moving detailed discussions on dataset construction to an appendix and provide a summary in the main manuscript.
    * Expand the related works section to include relevant literature on neural guided search and automated symbolic integration. Specifically, address connections to:
      * Neural Guided Search:
        * https://arxiv.org/abs/1906.04604
        * https://arxiv.org/pdf/2006.08381
        * https://arxiv.org/abs/2310.19791
        * https://arxiv.org/abs/2007.12101
        * https://arxiv.org/abs/2409.09359
      * Automated Symbolic Integration:
        * https://arxiv.org/pdf/2410.02666 (this is concurrent work).
    * Adding an overview paragraph at the beginning of the Experiments section would be helpful. This paragraph should ideally summarizes what research questions each experiment attempts to answer.
    * Table 2: The "Test set" and "Approach" column only list a single value. These columns are unnecessary. I recommend rewriting the caption to contain more information about relevant sections as well.
    * Table 3: Separate each approach into its own column for clarity.
    * Table 4: Clearly indicate the section and method relevant for the observations in this table.
    * L373: The algorithmic details should be moved to an algorithms section within Section 3.
    * Table 5: The symbolic baseline and the neural guided baseline should be explicitly pointed out.
    * L420: Alternating between the abbreviation "LC" and the full citation "Lample and Charton, 2020" is slightly awkward. Consider using only one of them.



`[1]`: https://arxiv.org/abs/1912.01412

`[2]`: https://www.cs.utexas.edu/~mooney/cs391L/paper-template.html

`[3]`: https://www.cs.williams.edu/~andrea/Carla/craft.html

`[4]`: https://deeplearning.cs.cmu.edu/F22/document/recitation/Recitation7/recitation_7_Paper_Writing_Workshop_F22.pdf

`[5]`: https://arxiv.org/pdf/1912.05752

---

### Official Review · Reviewer_EC72 · 2024-10-31

**Soundness:** 2
**Presentation:** 2
**Contribution:** 2
**Rating:** 5
**Confidence:** 4

**Summary:**

The paper presents a symbolic integration rules dataset comprising 27 million unique function and integration rule pairs. It introduces a new framework called guided integral steps to enhance symbolic integration. This framework combines a transformer model (LC[1], introduced in 2020) trained on the dataset with SymPy's integral_steps. The transformer predicts the most suitable rule based on the current expression tree. The authors demonstrate that the proposed method achieves improved accuracy and efficiency in integral prediction tasks.

**Strengths:**

1. The dataset offers valuable support to the relevant research community. The authors have also developed scripts to ensure the correctness of the generated steps.
2. The proposed method is intuitive and dynamically predicts the optimal next-step rule based on the current expression tree.
3. The experimental results in Table 4 demonstrate the method's effectiveness with limited nodes. The results in Table 5 further confirm the method's efficiency.

**Weaknesses:**

1. The paper offers limited technical contribution. It directly adopts the architecture and loss function from LC (introduced in 2020) without exploring more advanced model architectures or conducting comprehensive ablation studies on hyperparameters to validate the robustness of the proposed method.
2. The accuracy improvement reported in Table 3 is marginal compared to integral_steps and even shows a decline in performance on the BWD metric.
3. The paper’s writing could be improved. The scheme of guided_integral_steps is explained only through text, making it harder to follow. Visualizing the proposed method alongside previous ones, such as LC, in a figure or table would clarify the unique contributions of this work.

[1] Deep learning for symbolic mathematics

**Questions:**

1. Does "Number of Samples" in Tables 2 and 3 refer to the number of training samples? If so, why is the accuracy of the Substitution Rule significantly lower, despite having twice the number of samples compared to the Parts Rule?
2. The baseline model, LC, was proposed in 2020 and is somewhat outdated. Why didn't the authors explore more advanced methods to assess the robustness across different base models?

---

### Official Review · Reviewer_bSPQ · 2024-11-03

**Soundness:** 3
**Presentation:** 1
**Contribution:** 2
**Rating:** 3
**Confidence:** 4

**Summary:**

This paper proposes a deep learning-based method for indefinite integrals in mathematics. Namely, the authors propose a step-by-step approach that does not estimate the primal function for a function input as an integration target, but rather let it estimate the next integration rule to be applied. To do so, the authors collect a new synthetic dataset and train a Transformer-based model. The proposed approach is compared to the rule-based approaches of [Lample and Charton, 2020] and SymPy.

**Strengths:**

- It is an interesting and promising approach to have mathematical operations estimated step-by-step, rather than outputting the answer to the problem end-to-end.
- In this paper, the authors synthesize and publish a new dataset to successfully implement such an approach.
- The proposed method is compared to the closely related prior work [Lample and Charton, 2020] and the widely used SymPy, which shows the superiority of the proposed method on some metrics.

**Weaknesses:**

- In Figure 1, the motivation for why Depth First Search was adopted is not explained. The search method is thought to be an important factor that affects the efficiency of the proposed approach, and it seems that a discussion on what method should be selected and experimental verification were necessary.
- In Table 5, the proposed method is said to be three times more efficient than SymPy, but the definition of efficiency is not clear. The number of nodes searched is being compared here, but as the authors themselves acknowledge indirectly from line 438 onwards, SymPy is very fast to explore a single node because it is rule-based, and the proposed method is slow because it is based on deep learning. Ideally, a comparison of time and space complexity, as is being done only partially in the paragraphs from line 438 onwards, would be necessary, and the number of nodes searched does not seem to be a direct indicator of efficiency.
- On line 426, it says “Comparing to LC's FWD model helps demonstrate the OOD performance of guided integral steps using the BWD and IBP test sets.” However, since it is not quantitatively shown how different the SIRD-27M and FWD are from the BWD and IBP, respectively, it is not clear how fair the comparison between LC and the proposed method is.
- Furthermore, this manuscript is not sufficiently proofread as a technical paper.
  - Presentation issues:
    - In Figure 2, the right two columns each substitute different functions of x for u as the substitution rule, $u=e^{tan^{-1}(x)}$ and $u=tan^{-1}(x)$, but this is not explicitly written, and the presentation is confusing because the results of the integrals end with $u$ and $e^u$, which appear to be different from each other.
    - To the reviewer's recognition, in Table 3, integral_steps is a result with SymPy, LC is with [Lample and Charton, 2020], and only guided_integral_steps is the accuracy of the proposed method. In such cases, it was necessary to devise a way to clearly indicate which was the proposed method, such as placing the proposed method in the last line or adding some words such as (ours) to the proposed method name.
  - Trivial writing issues:
    - The sentence on line 276 starts with a lowercase letter.
    - For some reason, the rightmost column of the BWD row in Table 4 has a bold font in one place.
    - The format of each paper in the References section is a mess.

**Questions:**

- The reviewer would like to know the responses to Weaknesses.
- On line 323, it says “However, in the test set, we only have a single ground truth label for the function.” Does this mean that the test set is limited to samples with only one correct rule, or does it mean that, even though there are multiple correct rules, only one of them is considered to be the correct answer? If it is the former, the reviewer is concerned about the bias of the samples as a test set, and if it is the latter, concerned about the incompleteness of the evaluation method. Since it is a test set, shouldn't multiple correct rules be given as ground truth, even if it takes a labor effort?
- On line 370, it says “Consequently, we exclude such functions from the test sets of all three datasets before evaluation. Additionally, we remove functions with a sequence length greater than 384 when converted to prefix notation.” However, the reason for this is unclear. Even if there are problems that cannot be handled by the proposed method or the baseline, they should be included in the comparison. In particular, the restriction that series longer than 384 are not included is thought to work in favor of the proposed method, and the fairness of the comparison of methods is a concern.

---

### Official Review · Reviewer_fBDE · 2024-11-03

**Soundness:** 3
**Presentation:** 2
**Contribution:** 2
**Rating:** 3
**Confidence:** 3

**Summary:**

The paper presents SIRD-27M, a dataset of 27 million mathematical functions and their integration rules, and proposes using a transformer model to predict integration rules for symbolic integration. The model is integrated with SymPy's integral_steps function to create guided_integral_steps, which reduces the number of branches explored during integration by a factor of 3. The authors evaluate their approach on rule prediction and integral calculation tasks across multiple datasets.

**Strengths:**

The creation of a large-scale dataset (SIRD-27M) for predicting symbolic integration tasks (Complete Rule Prediction, Rule Prediction, and Integral Prediction) is valuable.

The integration with SymPy and the efficiency of  the proposed guided integral steps (explores 3× fewer branches than the original integral steps) demonstrates practical applicability.

It shows that a model trained on SIRD can guide the search for the integral, outperforming heuristics-based search.

**Weaknesses:**

Limited Technical Novelty:
The model architecture is a straightforward application of transformers for sequence prediction.
The approach of using neural networks to guide symbolic computation has been explored before (e.g., in AlphaGo and AlphaGeometry as mentioned in the paper).
No significant architectural innovations or novel learning techniques are proposed.

Significant Technical Limitations:
The model cannot handle hyperbolic trigonometric functions and their inverses.
Input sequences are limited to 384 tokens, restricting applicability to complex expressions.
The dataset is derived only from the FWD dataset, potentially limiting generalization.
The approach still relies heavily on SymPy's underlying integration capabilities.

Incomplete Evaluation:
No comparison with current state-of-the-art large language models (e.g., GPT-4, Claude Sonnet 3.5). It is possible SOTA models can largely solve the rule prediction problem.
Limited analysis of failure cases and error patterns.

Methodological Concerns:
Limited discussion of the model's generalization capabilities beyond the test sets

**Questions:**

Have you evaluated how modern LLMs like GPT-4, Sonnet3.5 perform on your dataset?

How does the model perform on real-world integration problems from mathematics textbooks or research papers?

How could you address the limitations of model can not handling hyperbolic functions and their inverses?

The paper presents a well-executed but incremental contribution to symbolic integration. While the dataset creation and integration with SymPy are valuable, the technical approach lacks novelty and has significant limitations. The absence of comparisons with modern LLMs and the restricted handling of mathematical functions make it difficult to assess the true value of this contribution. I recommend rejection but encourage the authors to address these limitations in future work.

---

### Meta-Review · Area_Chair_RwSt · 2024-12-20

**Metareview:**

This paper introduces a new dataset (SIRD-27) for studying symbolic integration rules (27 million samples over 24 rules). The authors train a transformer model over this dataset, where given a function as input, the model predicts the next integration rule to apply (the predicted integration rule is then applied by SymPy). The model is evaluated on the accuracy of the next rule prediction, as well as the end-to-end integral solving task, and results show it performs better than a baseline (Lample & Charton (2020)). While the reviewers appreciate the problem the authors are tackling, and the proposed step-by-step approach, there remains key concerns on the baseline choices (e.g. those suggested by reviewer qgw3), limitations on the technical approach to handle some of the integration rules, and additional evaluations to consider (other datasets and analysis to understand points of failure and gaps). There are also various presentation issues pointed out by reviewers that authors are encouraged to address (sample works from reviewer qgw3 may be especially helpful for the authors to reference for future organization and formatting).

**Additional Comments On Reviewer Discussion:**

All reviewers initially voted to reject the paper and there were no response from authors during the rebuttal period.

---

### Decision · Program_Chairs · 2025-01-22

Reject